# Profile of Bioactive Components and Antioxidant Activity of *Aronia melanocarpa* Fruits at Various Stages of Their Growth, Using Chemometric Methods

**DOI:** 10.3390/antiox13040462

**Published:** 2024-04-14

**Authors:** Natalia Dobros, Agnieszka Zielińska, Paweł Siudem, Katarzyna Dorota Zawada, Katarzyna Paradowska

**Affiliations:** Department of Organic and Physical Chemistry, Faculty of Pharmacy, Medical University of Warsaw, Banacha 1, 02-097 Warsaw, Poland; natalia.dobros@wum.edu.pl (N.D.); pawel.siudem@wum.edu.pl (P.S.); katarzyna.zawada@wum.edu.pl (K.D.Z.); katarzyna.paradowska@wum.edu.pl (K.P.)

**Keywords:** *Aronia melanocarpa*, chokeberry, CGAs, anthocyanins, antioxidant activity, chemometrics, PCA

## Abstract

Aronia (chokeberry, *Aronia melanocarpa*) is a valuable fruit that combines the health benefits of green tea and red wine and is gaining popularity worldwide. Aronia has a unique chemical composition with strong antioxidant properties, including anthocyanins and chlorogenic acids (CGAs). However, it remains unclear which specific compounds or groups are primarily responsible for the antioxidant properties of chokeberry. Therefore, an analysis of the antioxidant properties of aronia berries based on geographical region and their variability during ripening (from green to frostbitten fruit) was conducted. The fruits were collected from three locations for two years. The aim of our work was to identify the compounds responsible for the antioxidant properties of aronia berry extracts by using chemometric methods. The analyses of anthocyanins and CGAs were performed using HPLC-DAD, and the antioxidant capacity was assessed by FRAP and DPPH methods. The PCA analysis also considered variations in temperature and precipitation. The chemometric analysis revealed a strong correlation between radical-scavenging properties and the content levels of chlorogenic acids. The results obtained in this study show that unripe green chokeberry fruits exhibit the highest antioxidant properties, which can be attributed to the high content of CGAs at this stage.

## 1. Introduction

The potential sources of antioxidant phenolic compounds have been extensively studied in various types of plant material. This includes vegetables, fruits, herbs and spices, leaves, bark, roots, seeds (including oilseeds), and cereal plants. Fruits and vegetables are crucial components of our daily diet as they are low-calorie raw food materials and rich sources of carbohydrates, including fiber. They also provide essential minerals and vitamins that regulate metabolic processes in the human body. Moreover, they contain polyphenolic compounds that protect our bodies against oxidative stress [1]. *Aronia melanocarpa* (Michx.) Ell., known as chokeberry or aronia, is a shrub belonging to the *Rosaceae* family, widely distributed in Europe and around the world, and its berries contain an extensive range of polyphenols with antioxidant properties. Berries, including chokeberry, owe their biological activity mainly to anthocyanins, which are more present in chokeberry than in elderberry or black currant. Due to these reasons, chokeberry is often referred to as a superfruit [2].

Chokeberry fruit is primarily used to produce syrups, juices, and wine. Recently, due to the well-documented health benefits of the polyphenolic compounds found in chokeberry fruits, there have been efforts to utilize berries and their by-products for food enrichment and as a dietary supplement. The consumer market shows a keen interest in antioxidants, and chokeberries are well-positioned to compete with citrus fruits. Including superfruits in our diet and food production can positively impact human physical and mental health and development. Numerous scientific reports confirm the high antioxidant activity of dark berries, including *A. melanocarpa* [3,4,5,6]. Ripe chokeberries are rich in anthocyanins and have numerous health benefits. They exhibit anti-inflammatory and antibacterial properties, reduce the risk of cardiovascular diseases, and have anti-diabetic and anti-cancer potential [7,8,9,10,11]. The biological activity of berries may be related to their antioxidant properties [12]. The effectiveness of antioxidants found in plants may vary based on factors such as the plant’s maturity, climate, and variety. As a result, it is important to conduct both qualitative and quantitative tests on plant materials, including the testing of chokeberry fruits grown in different latitudes. Additionally, monitoring the levels of active substances, such as antioxidants, is important during fruit growth [13]. The evaluation of samples with complex chemical composition, such as chokeberry fruit, poses a challenge for present analytical techniques. These techniques are a source of complex analytical signals, the so-called fingerprint methods. Chemometric methods aid in exploring, classifying, and interpreting large data matrices.

The aim of the study was to characterize the chemical composition of *A. melanocarpa* fruits, depending on their stage of growth and place of harvest, and to determine how these factors affect the fruit’s antioxidant activity. The chokeberry samples were gathered from three distinct regions of Poland and over two consecutive seasons. To establish a correlation between the sample’s content and properties, principal component analysis (PCA) was employed, a method which has been utilized in other studies [14,15]. An assessment of PCA was conducted among all the tested samples and variables. The variables include the total content of polyphenols and flavonoids, which were measured using spectrophotometric methods. Also, the content levels of chlorogenic acids, chlorogenic (CGA) and neochlorogenic (nCGA), and the anthocyanins cyanidin-3-O-galactoside (CyaGal) and cyanidin-3-O-arabinoside (CyaAra) were measured using HPLC-DAD. The antioxidant activity was tested using the DPPH radical-scavenging test and ferric ion-reducing antioxidant power (FRAP) assay. Our results can help eliminate errors that arise when comparing research data, including differences in composition, collection time, and growth location.

## 2. Materials and Methods

### 2.1. Plant Material

*Aronia melanocarpa* (Michx.) Elliott berries were collected over two seasons extending from May to October of 2016 and 2017. The chokeberry fruits were gathered from three different locations in Poland, and the samples were taken at two-week intervals (Table 1). The first farm is situated in the central region of Poland, specifically in the Mazovian Voivodeship, Grójec. This region experiences a relatively large number of warm and cloudy days, averaging 63 such days per year. Days with very warm and cloudy weather without precipitation are prevalent. The average annual rainfall is 540 mm, and the average annual temperature is +7.7 °C. The hottest month is July, with an average temperature of +18.9 °C, whereas the coldest month is January (av. −3.6 °C) [16]. The second farm is located in the western part of Poland, in the Greater Poland Voivodeship. This region belongs to the temperate climate zone, where marine and continental influences intersect, and has a milder climate than found on the first farm. The average annual temperature is around +8.5 °C, with a small number of frosty days throughout the year and low rainfall, which amounts to 500–550 mm, one of the lowest in Poland. This region is characterized by weaker winds than found in Poland’s northern and southern parts [17]. The third farm is located in southeastern Poland, in the Lubelskie Voivodeship. Influenced by both continental and oceanic air masses, this region is characterized by long, sunny summers, frequently cold winters, and moderate rainfall. The average annual temperature is about +7.5 °C [18]. Table 1 includes the fruit characteristics in terms of ripeness and color, as well as the average rainfall and altitude of the farms. Detailed harvest dates and associated sample numbering are shown in Table 2. 

The fruit specimens were collected randomly from the healthy bushes in the middle of a field and frozen until the extraction procedure. The voucher specimens were deposited at the Herbarium of the Department of Biology and Pharmaceutical Botany (Medical University of Gdansk, Poland).

### 2.2. Extraction Procedure

The extraction process was performed according to the procedure described by Oszmiański and Wojdyło [1]. Fresh fruits were soaked in hexane to remove the wax layer from the peel; freeze-dried by use of Alpha 1-2 LDplus CHRIST lyophilizer (Martin Christ, Osterode am Harz, Germany) at −25 °C, 0.63 mbar, for 168 h; and finally ground into powder. Then, one gram of plant material was mixed with 30 mL of methanol acidified to pH 1.0 and sonicated for 20 min at room temperature. The obtained extracts were centrifuged at 3000 rpm for 10 min and then filtered through Whatman filter paper. All samples were prepared in triplicate. One part of the obtained extracts was stored at −32 °C until the total polyphenols, total flavonoids, and antioxidant analyses were conducted. The second part of the extracts was concentrated with a rotary evaporator (Heidolph Instruments, Schwabach, Germany) and then freeze-dried at −25 °C, 0.63 mbar for 120 h. The obtained samples were stored in a refrigerator until further analysis.

### 2.3. Total Phenolic Content

Total polyphenols were determined according to the Folin–Ciocalteu colorimetric procedure previously described by Waterhouse [19]. Firstly, 1185 μL of demineralized water (MilliQ^®^, Merck Millipore, Burlington, MA, USA) was added to 15 μL of chokeberry extract or gallic acid solution, and the resulting mixture was mixed with 75 μL of Folin–Ciocalteu reagent and 225 μL of 20% Na_2_CO_3_. After twenty minutes of thermostatting at 40 °C the absorbance was measured at 765 nm using a UV-Vis spectrophotometer (Evolution 60S, Thermo Scientific, Waltham, MA, USA). Each analysis was carried out in triplicate. The total polyphenol content was calculated as gallic acid equivalents (GAE) according to the standard curve: Absorbance = 0.0011 × C [mg GAE/mL] + 0.0016, R^2^ = 0.999 (mg/L). The final results were expressed per 1 g dry weight (mg GAE/g DM).

### 2.4. Total Flavonoid Content

The total flavonoid content test was performed using the modified colorimetric procedure described by Vábková [20]. Firstly, 100 μL of chokeberry extract or a catechin solution was diluted with 1400 μL of demineralized water, and subsequently mixed with 60 μL of 5% NaNO_2_ and 60 μL of 10% AlCl_3_. A total of 400 μL of 1 M NaOH was added after five minutes of thermostatting at 25 °C, and then the absorbance was measured at 510 nm using a UV-Vis spectrophotometer (Evolution 60S, Thermo Scientific, Waltham, MA, USA). Each analysis was carried out in triplicate. The total flavonoid content was calculated as catechin equivalents (CE) according to the standard curve: Absorbance = 0.0016 × C [mg CE/L] + 0.0053, R^2^ = 0.998 (mg/L). The final results were expressed per 1 g dry weight (mg CE/g DM).

### 2.5. The Ferric Reducing Antioxidant Power (FRAP Assay)

The ferric-reducing antioxidant power assay was performed using the modified method described by Benzie and Strain [21]. Briefly, 1000 μL of FRAP reagent (300 mM acetate buffer with pH 3.6, 10 mM TPTZ solution and 20 mM FeCl_3_, in a ratio of 10:1:1) was added to 50 μL of chokeberry extract or Trolox solution. The absorbance was measured at 593 nm using a UV-Vis spectrophotometer (Evolution 60S, Thermo Scientific, Waltham, MA, USA) after four minutes of thermostatting at 37 °C. Each analysis was carried out in triplicate. The ferric reducing antioxidant power was calculated as Trolox equivalents (μM/g d.m.) based on the standard curve: Absorbance = 1.0022 × C [µm Trolox/L] + 0.2491, R^2^ = 0.998 (μM/L). The final results were expressed per 1 g of dry weight (µm Trolox/g DM).

### 2.6. DPPH Radical-Scavenging Assay

The radical-scavenging properties were assessed with the 2,2-diphenyl-1-picrylhydrazyl (DPPH) assay based on the work of Sanna et al. [22]. Equal volumes of a methanol-diluted (30–100-fold) aronia extract and DPPH methanolic solution (3.4 mmol/L) were mixed, and after 20 min EPR (electron paramagnetic resonance) spectra were registered. A blank sample with methanol instead of an extract was also prepared for each run. EPR spectra were measured at room temperature (298 K) on a MiniScope MS200 EPR spectrometer (Magnettech, Freiberg, Germany). The intensity of the obtained EPR spectra were compared with the blank. The results were expressed as Trolox equivalents (TE) in micromoles of Trolox per gram of dry matter of plant material.

### 2.7. HPLC Analysis

Qualitative and quantitative analyses were performed by HPLC-DAD. Anthocyanin and chlorogenic acids profiles were characterized simultaneously using the Hitachi Chromaster system (Hitachi High-Tech, Tokyo, Japan) with the use of a Purospher STAR (Merck KGaA, Darmstadt, Germany) RP-18e column (5 µm, 4 × 250 mm), according to methods validated and described in our previous publication on the analysis of chokeberry extracts [23]. A mobile phase gradient system consisted of 4.5% (*v*/*v*) formic acid (A) and acetonitrile (B). The gradient conditions were as follows: 0–5 min (5% B), 5–15 min (5–8% B), 15–50 min (8–25% B), 50–55 min (25–50% B), and 55–65 min (5% B). Flow rates were as follows: 1–15 min, 1 mL/min; 15–50 min, 0.8 mL/min; and 50–65 min, 1 mL/min. The CGAs chromatograms were monitored at 330 nm and the anthocyanin at 520 nm. Retention times were as follows: neochlorogenic acid (nCGA)—10.9 min, chlorogenic acid (CGA)—20.5 min, cyanidin 3-galactoside (CyaGal)—26.8 min, and CyaAra—cyanidin 3-arabinoside (CyaAra)—30.7 min. All measurements were performed in triplicate. The concentrations of the compounds were determined using an appropriate calibration curve. The final results were expressed per 1 g of dry weight.

### 2.8. PCA Analysis

PCA was carried out using Statistica 10 (StatSoft Inc., Tulsa, OK, USA). PCA was used to visualize and overview the data and reduce the number of variables. Prior to PCA, all data were normalized using the Z-score function. The results of the analysis were displayed visually as a scores plot. The plots were used to observe grouping in the data sets. One-way analysis of variance (ANOVA) was applied, along with the Tukey test for significance, *p* < 0.05.

## 3. Results and Discussion

### 3.1. HPLC Analysis of Chlorogenic Acids and Anthocyanins

The DAD-HPLC method was used to determine the content of individual anthocyanins (cyanidin 3-galactoside CyaGal, cyanidin 3-arabinoside CyaAra) and chlorogenic acids (neochlorogenic acid nCGA and chlorogenic acid CGA) in the extracts from chokeberry fruit. All the measurement data obtained are included in Table 2. Figure 1 illustrates the changes in these contents during the growing season across all farms. In the first year of sampling, no significant differences were observed among individual farms (as shown in Figure 1a). The average content ratio of 3-galactoside (CyaGal) to cyanidin 3-arabinoside (CyaAra) was approximately 1.40 ± 0.20, with more fluctuations observed during the first year of sampling. Aronia berries also contained significant amounts of neochlorogenic and chlorogenic acids, with an average ratio of nCGA to CGA equal to 0.94 ± 0.25.

The changes in the content levels of the chlorogenic acids and anthocyanins followed a characteristic pattern. The highest content was observed for chlorogenic acids at the beginning of the harvest, with a concentration of 15–20 mg/g DW. This was followed by a decrease, which coincided with an increase in anthocyanin content in mid-July. This particular pattern may be due to the function of chlorogenic acids as defense agents against insects. This function is essential during the fruit growth stage [24]. Additionally, CGA may act as a precursor for synthesizing anthocyanins, which is discussed in more detail in Section 3.2. Ripe chokeberries become rich in anthocyanins, pigments that give ripe berries red, blue, and purple colors. As berries ripen, anthocyanin content increases dramatically, providing a visual cue to distinguish early from fully ripe berries.

In the second year, a longer period was observed during which the samples were characterized by a high content of anthocyanins (as shown in Figure 1). This was likely related to the extension of the growing season across all farms. When comparing the weather data for Farm 1 (as seen in Figure 2), it is visible that during September of the second year, there was significantly more precipitation despite similar temperature patterns. During dry periods, the plants tend to shed their fruits and leaves more quickly, accelerating and shortening the growing season.

### 3.2. The Total Polyphenol and Total Flavonoid Content Levels in Aronia Extracts

The list of plant secondary metabolites includes phenolic compounds, and their presence can be easily assessed by measuring total phenolic (TP) and total flavonoid (TF) content levels. These results take into account the entire range of polyphenol and flavonoid compounds present in the sample and constitute a good measure of the variability of the content of these ingredients. Aronia also contains derivatives of quercetin, proanthocyanins, and other phenolic acids, although in smaller quantities. Thus, the designations TP and TF allow us to consider their presence in our analyses.

Figure 3 displays the changes in total phenolics and flavonoids in samples taken from three locations over a year. Extracts from fruits in the initial growth phase (green fruit) are characterized by higher contents of phenolic compounds and flavonoids (the highest amounts being 130 ± 5 mg GAE/100 g and 115 ± 6 mg CE/100 g) than fruits in the later ripening phase. This change occurs in mid-July and is associated with a decrease in chlorogenic acid content and the appearance of anthocyanins. Furthermore, the ripening process of the fruit from unripe to fully ripe results in a significant reduction of total phenolics (TP) and flavonoids (TF). Interestingly, flavonoids constitute 75% of the total polyphenolics. These findings suggest that green fruits can be considered a potential source of natural phenols and flavonoids (along with chlorogenic acid) for pharmaceutical and food purposes. Additionally, the phenolics, which are known for their antioxidant activity, reflect the changes in the antioxidant properties. As the antioxidant capacity decreases, there is a significant decrease in antioxidant capacity (as shown in Figure 4 and Table 3). Moreover, the reduction in the content level of polyphenols, which are known for their antioxidant properties, during fruit ripening corresponds to the decrease in antioxidant activity observed at the same time (Figure 4).

The pattern of decreasing non-anthocyanin polyphenols and increasing anthocyanins during fruit growth is typical and has also been observed in other berries like blueberry [25] and strawberry [26]. Kalt et al. [27] examined the composition and antioxidant activity of highbush blueberry fruits at different stages of ripeness and during storage. The chemical content of blueberries is similar to that of chokeberry, with anthocyanins and chlorogenic acid being the main components, along with other flavonoids like catechin and proanthocyanidin. The decline of total phenolics as anthocyanins increase was observed and explained by the fact that phenolics are used for anthocyanin synthesis. The study has shown that the amounts of anthocyanins formed during the storage of fruits depend on their ripeness at the time of harvest. Unripe fruits have lower anthocyanin synthesis compared to ripe ones. The increase in anthocyanin content after harvest is mainly due to the conversion into anthocyanins of non-anthocyanin phenolic precursors, such as chlorogenic acid, which are present in the fruits. It is worth noting that not all fruits experience a decline in phenolic content during ripening. For instance, raspberries have substantially higher levels of total phenolics in red ripe fruits compared to pink underripe fruits [28].

### 3.3. Antioxidant Properties

The antioxidant properties of aronia fruits were evaluated using DPPH radical and FRAP assays to analyze radical-scavenging properties and reducing properties, respectively. The changes in both these properties throughout the growth and maturation season are depicted in Figure 4, and all results have been compiled in Table 3. 

The DPPH method revealed that the radical-scavenging activity was approximately twice as high for unripe *A. melanocarpa* fruits collected at the initial stages of growth, compared to the ripe ones. There is a statistically significant difference in the radical-scavenging activity of unripe and ripe aronia fruits across all places of origin and in both years of collection. Fruits collected from the end of May until the beginning of July, i.e., before ripening, exhibited the highest radical-scavenging activity. This finding aligns with the results of Yang et al. for *A. melanocarpa* cultivated in South Korea [29]. At the end of July (24.07), activity significantly decreased when ripening began, and, after a minor increase, remained approximately constant from August until October. An exception was observed for fruits from Farm 2 collected in 2017, where the decrease in radical-scavenging activities was noted later, on 7 August 2017. This pattern of changes in antioxidant properties during fruit development and maturation corresponds with the changes in chlorogenic acid content but not the anthocyanin content, indicating the significant role of chlorogenic acids in the radical scavenging of aronia fruits.

For the reducing properties, as determined with the FRAP assay, a much smaller difference was observed between unripe and ripe fruits. This is due to the much lower values obtained for unripe fruits in the FRAP assay compared to the DPPH assay. However, the pattern is similar—higher values were generally obtained for unripe fruits, both in 2016 and 2017, for all farms where the fruits were collected throughout the entire season. The only exception was observed in the case of Farm 2 (F2), where the fruits collected on 21 August 2016 exhibited reducing properties similar to those of the unripe ones. This difference was more prominent for fruits collected in 2017. However, the minimum observed value for fruits collected on 24.07 was much more notable for 2016.

The decrease with ripening in the reducing properties of chokeberry, as determined with the FRAP assay, was observed previously for aronia by Sosnowska et al. [30], though only four points of fruit maturation were studied. A similar effect was observed for the Cristalina variety of sweet cherry, and it was ascribed to oxidative stress due to increased metabolism [31]. The distinctly higher radical-scavenging and reducing properties, as well as the higher content levels of CGAs (Table 2), total polyphenols, and total flavonoids (Table 3) in fruits collected up to the beginning of July might be due to the higher rates of synthesis of secondary metabolites to protect the fruits against environmental stressors (e.g., UV radiation), which induce oxidative stress, and against herbivores, including insects [32,33], or to the increased metabolism due to the presence of chloroplasts present in green fruits, which are responsible for photosynthesis, driving the central metabolism and development.

The decreases in CGAs, radical-scavenging activity, and reducing capacity at the onset of fruit ripening could be attributed to the peak accumulation of radical oxygen and nitrogen species (RONS) [34]. This hypothesis is supported by the earlier-reported significant decrease in the procyanidin content of aronia berries at the start of ripening [35]. Furthermore, it seems that anthocyanins, which begin to form at this time and have been demonstrated to help mitigate the effects of high-light stress [36], act mainly as photoprotection agents rather than radical scavengers since their appearance does not compensate for the loss of early-stage antioxidants, despite their well-established radical-scavenging activity. 

### 3.4. PCA Analysis

PCA analysis was performed on the data obtained from the analysis of chokeberry extracts. A PCA plot is shown in Figure 5, where the first principal component (PC1) describes 58.15%, and the second (PC2) 21.15% of the total variance. Almost 80% of the total variance is explained by the first two PCs; therefore, only PC1 and PC2 were taken for further analysis.

The score plot reveals two distinct groups. The samples from May or June up to mid-July are clustered on the right side of the graph (marked with an ellipse). These samples correspond to the period when the plant has the highest content levels of chlorogenic acids (CGA and nCGA), total flavonoids, and polyphenols. The presence of these compounds significantly impacts the high results of the antioxidant test using the DPPH radical.

On the left side of the graph, the points relating to the extracts obtained from the months from mid-July to October are grouped (marked with an ellipse). These points correspond to fruits with the highest anthocyanin content. PC1 is a new variable that corresponds to the division of fruit maturity into two periods. Regardless of the analyzed year or geographical location of the plantation, there is always a rapid change in the profile of chokeberry fruits in mid-July—a decrease in the content of chlorogenic acids and an increase in the amount of anthocyanins. The farm location had no significant impact on the bioactive compound content levels and their antioxidant properties. On the other hand, the PC2 variable corresponds to weather conditions—temperature and precipitation. Maintaining an appropriate temperature (approx. 18–19 °C) causes a change in the profile, which results in the location of points on the right or left side of the graph. Interestingly, the FRAP test is sensitive to high levels of both chlorogenic and anthocyanins.

Analyzing the content levels of polyphenols and flavonoids during the ripening of chokeberry fruits shows that 70% of polyphenols are compounds from the flavonoid group. They are most abundant in the initial phase of fruit ripening (green fruit), in which the highest content of chlorogenic acids is also observed. At this ripening stage, they are mainly responsible for antioxidant properties and protective functions. With ripening, the antioxidant functions are taken over by anthocyanins, the amount of which increases. However, fruits with the highest anthocyanin content do not have the highest antioxidant properties. The high content of chlorogenic acids in the initial phase of growth has a dominant influence on the high results obtained in antioxidant tests. It seems that the research findings suggest that chokeberry extracts can be optimized to suit specific desired properties. For instance, if better antioxidant properties are desired, it is recommended to use green fruits that are rich in chlorogenic acids. This is according to the points on the right side of the plot in Figure 5. On the other hand, if one needs an extract with anti-inflammatory and antibacterial properties, ripe fruits (purple and black) that are rich in anthocyanins can be used. However, it is worth noting that such extracts exhibit lower antioxidant properties, as indicated by the points on the left side of the plot in Figure 5. In previous studies of cactus pear fruit, a correlation of the measured amounts of polyphenols and ascorbic acid with the result of the DPPH test was observed [37,38]. In the case of aronia fruits, which are not a rich source of vitamin C, the grouping is based on the content levels of chlorogenic acids and total polyphenols, and high results in the DPPH test. In methua fruit (*Madhuca longifolia*), which is primarily a source of flavonoids when ripe, clustering by the new variable PC into ripe and unripe fruit was also observed [39]. Similarly, Gomez et al. [40] and Rajkumar et al. [41] described the study of the Chinese variety of Satsuma mandarin (*C. reticulata*) and banana fruits, respectively, based on their degrees of ripeness, using PCA analysis. Also, in pomegranate fruits, five stages of ripening grouped according to increasing levels of cyanidin glucosides were observed [42]. As in the case of chokeberry, unripe fruits were grouped according to high DPPH test results. For chokeberries, it is associated with a high content of chlorogenic acids, and for pomegranate fruits with other phenolic acids (e.g., gallic and ellagic).

The results of our research indicate a significant impact of the presence of chlorogenic acids on the antioxidant properties of fruits, regardless of their harvest time. However, it also suggests that further investigation into the health benefits of unripe chokeberry fruits is necessary, focusing on an extract containing chlorogenic acid as a key research compound. Chlorogenic acid is a compound that may be beneficial for people with diabetes, according to early in vivo studies conducted on obese rats with genetic insulin resistance [43]. Its administration to this group significantly reduced cholesterol concentration (44%) and triacylglycerols (60%) in the blood serum and liver. Furthermore, the rats gained less weight compared to the control group, and a decrease in blood glucose concentration after a meal was observed. Many studies indicate the relationship between the strong antioxidant properties of compounds such as flavonoids and phenolic acids and their roles in alleviating the symptoms of diabetes [12,44]. For this reason, due to the ability to select chokeberry fruits (both ripe and unripe) with the highest antioxidant properties, it is possible to incorporate them into the formulation of functional food ingredients and dietary supplements with desirable health-promoting properties. 

## 4. Conclusions

Ripe chokeberry fruits have been the subject of numerous chemical and biological analyses over the last years. However, there is still insufficient information on the chemical compositions and effects of the fruits as varying during their growth, particularly for unripe and semi-ripe ones.

Phenolic compounds play critical roles in plant growth and development, particularly in defense mechanisms. They exhibit potent antioxidant properties, effectively counteracting the impact of oxidative stress. The results obtained in this study suggest that unripe green chokeberry fruits also contain nutrients and antioxidants and exhibit antioxidant activity. The content levels of chlorogenic acids, total polyphenols, and total flavonoids in fruits harvested until the beginning of July are significantly higher, which is also reflected in the antioxidant properties. However, ripe fruits were characterized primarily by a higher anthocyanin content. The level of anthocyanins in the fruit increases as it ripens. However, the fruit in the initial growth phase has the highest antioxidant properties. Therefore, the high content of chlorogenic acids has a dominant influence on the high antioxidant properties. This information may be useful for obtaining chokeberry extracts with the desired properties.

In addition to classical statistical methods, multivariate analysis methods are used to determine correlations between individual groups of compounds and their properties and to enable data visualization, such as in the form of dendrograms. Among these methods, principal component analysis (PCA) is frequently used. The study of extracts prepared from chokeberry fruits collected at eleven stages of growth allowed for the determination of differences in the content of polyphenolic compounds, both quantitatively and qualitatively. A relationship between the composition and antioxidant properties of the analyzed extract was also demonstrated.

## Figures and Tables

**Figure 1 antioxidants-13-00462-f001:**
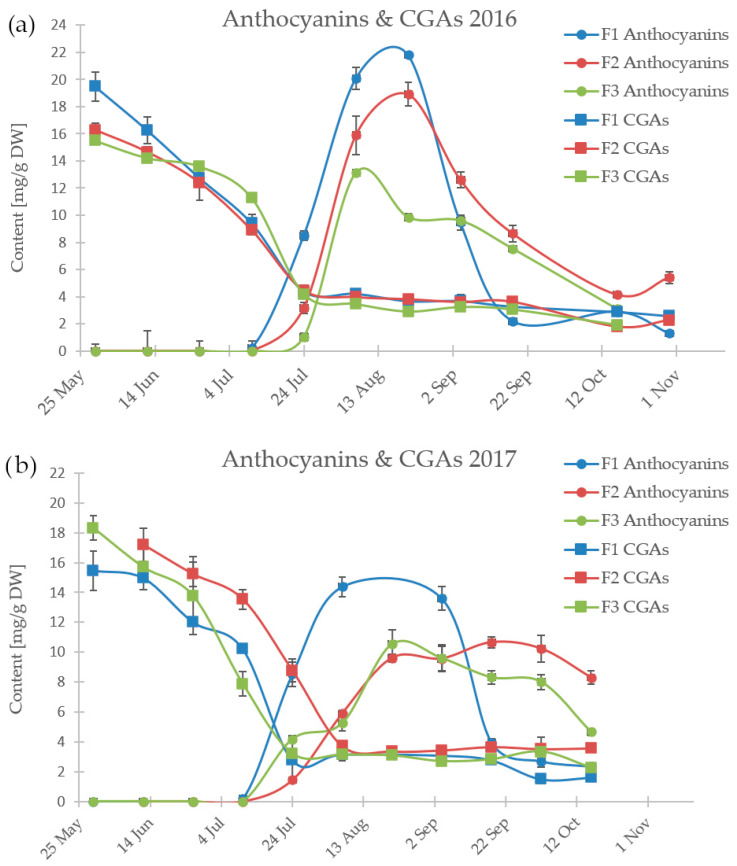
The changes in the CGA and anthocyanin contents of fruits harvested between May and October from three locations for two years ((**a**) 2016, (**b**) 2017). Abbreviations: Anthocyanins (the sum of CyaGal and CyaAra content), CGAs (the sum of nCGA and CGA); both as measured by HPLC-DAD.

**Figure 2 antioxidants-13-00462-f002:**
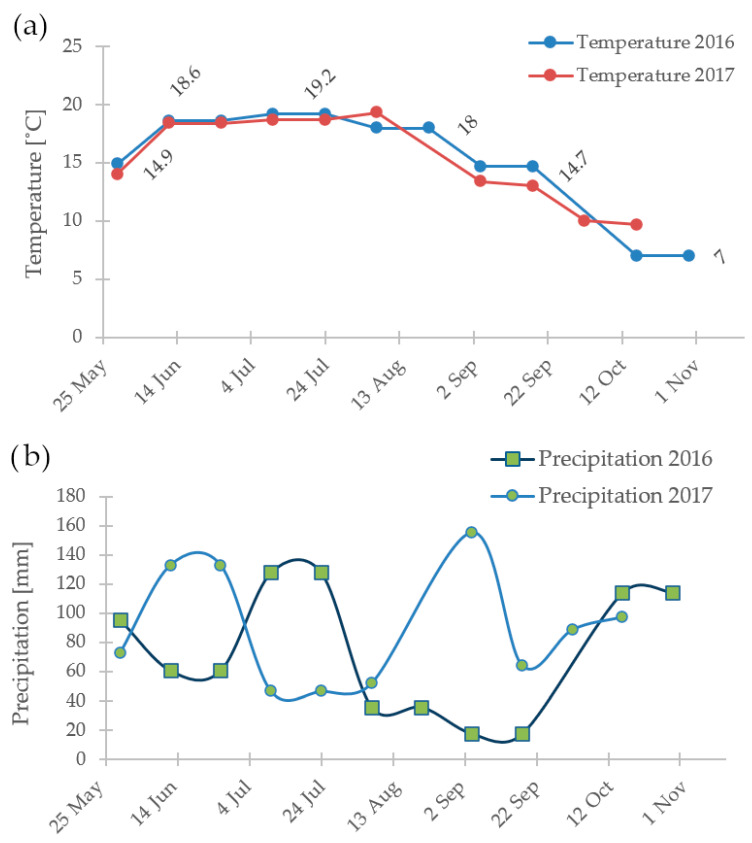
The yearly differences in temperature (**a**) and precipitation (**b**) for Farm 1 over two years.

**Figure 3 antioxidants-13-00462-f003:**
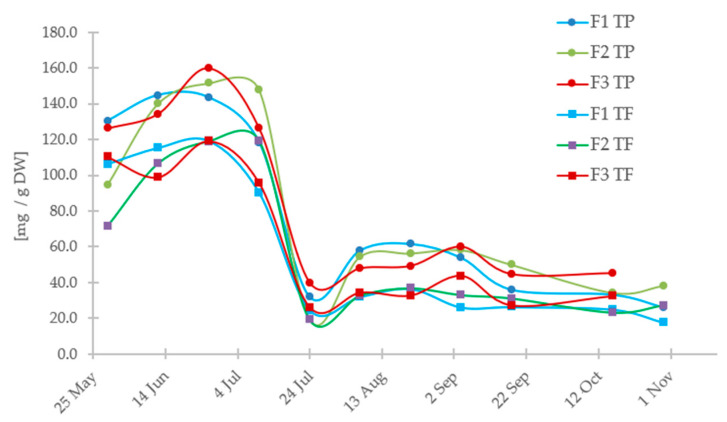
The total content levels of polyphenols (TP, mg GAE/g DW) and flavonoids (TF, mg CA/g DW) of berries collected in 2016.

**Figure 4 antioxidants-13-00462-f004:**
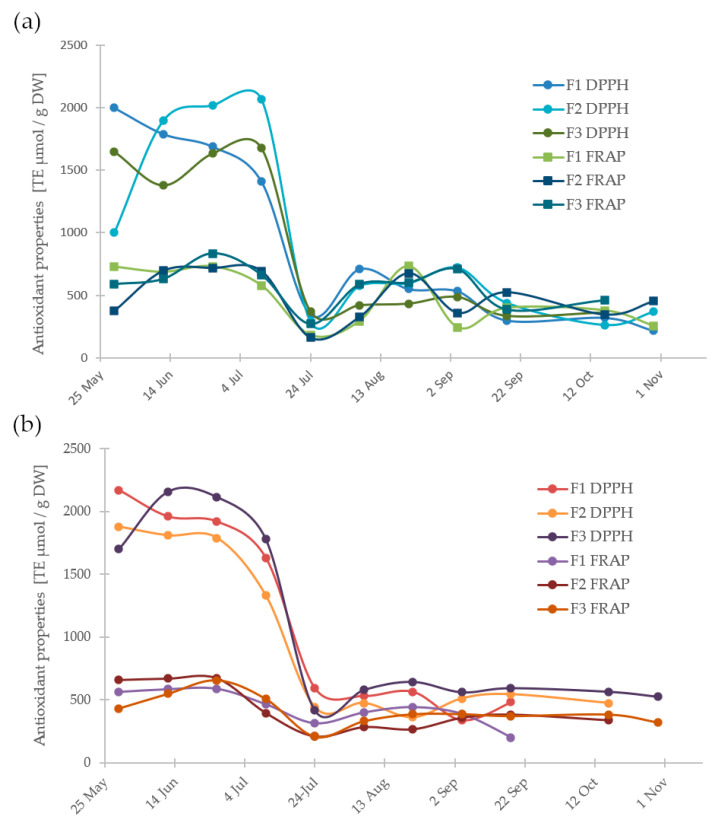
The changes in the antioxidant properties, as determined with DPPH (circles) and FRAP (squares) assays; samples taken from May to October from three locations for two years: (**a**) 2016, and (**b**) 2017.

**Figure 5 antioxidants-13-00462-f005:**
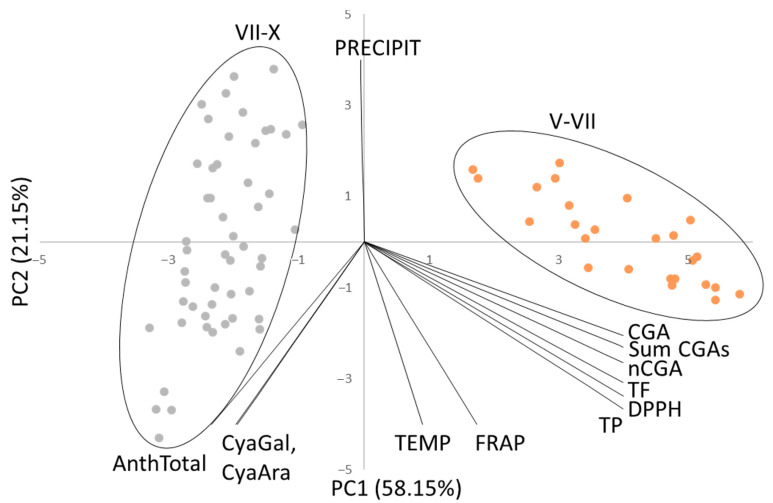
Score plots of principal component analysis of chokeberry extract (PC1 versus PC2 score).

**Table 1 antioxidants-13-00462-t001:** Characteristics of the farms and aronia berry samples. AMSL—above mean sea level.

	Average Temp [°C]	Altitude (AMSL) [m]		29 May– 10 July	24 July– 18 September	October
Farm 1	+7.7	98	Color	Green	Red/Dark red	Black/Purple
Farm 2	+8.5	82	Maturity stages	Unripe	Ripe	Overripe
Farm 3	+7.5	180				

**Table 2 antioxidants-13-00462-t002:** The content levels of anthocyanins and chlorogenic acids in chokeberry fruits were obtained by HPLC-DAD analysis [mg compound/1 g of DW]. Different letters mean statistical difference by Tukey’s test (*p* < 0.05). Abbreviations and definitions: Farm 1 (F1.x), Farm 2 (F2.x), Farm 3 (F3.x); CyaAra (cyanidin 3-arabinoside); CyaGal (cyanidin 3-galactoside); Sum of Anthocyanins (sum of CyaGal and CyaAra); nCGA (neochlorogenic acid); CGA (chlorogenic acid); Sum of CGAs (sum of nCGA and CGA); ND—not detected.

Sample No.	Date of Collection	CyaAra	CyaGal	Sum of Anthocyanins	nCGA	CGA	Sum of CGAs
	2016						
F1.1	29 May	ND	ND	ND	9.75 ^a^ ± 0.55	9.74 ^a^ ± 0.53	19.49 ^a^ ± 1.08
F1.2	12 June	ND	ND	ND	9.30 ^a^ ± 0.52	6.96 ^b^ ± 0.48	16.26 ^b^ ± 1.00
F1.3	26 June	ND	ND	ND	6.73 ^b^ ± 0.2	6.02 ^c^ ± 0.16	12.75 ^c^ ± 0.36
F1.4	10 July	0.14 ^a^ ± 0.01	0.05 ^a^ ± 0.01	0.20 ^a^ ± 0.02	4.81 ^c^ ± 0.42	4.6 ^d^ ± 0.25	9.41 ^d^ ± 0.67
F1.5	24 July	4.68 ^b^ ± 0.21	3.27 ^b^ ± 0.13	8.50 ^b^ ± 0.34	2.03 ^d^ ± 0.1	2.36 ^e^ ± 0.11	4.39 ^e^ ± 0.21
F1.6	7 August	12.54 ^c^ ± 0.5	7.55 ^c^ ± 0.33	20.09 ^c^ ± 0.83	1.83 ^d^ ± 0.08	2.39 ^e^ ± 0.09	4.21 ^e^ ± 0.17
F1.7	21 August	13.4 ^d^ ± 0.14	8.41 ^d^ ± 0.07	21.8 ^d^ ± 0.21	1.48 ^e^ ± 0.02	2.18 ^f^ ± 0.04	3.66 ^f^ ± 0.06
F1.8	4 September	5.09 ^e^ ± 0.33	4.52 ^e^ ± 0.22	9.48 ^e^ ± 0.55	1.48 ^e^ ± 0.22	2.19 ^f^ ± 0.26	3.67 ^f^ ± 0.48
F1.9	18 September	1.16 ^f^ ± 0.12	0.91 ^f^ ± 0.10	2.19 ^f^ ± 0.22	1.27 ^f^ ± 0.08	2.00 ^f,g^ ± 0.12	3.26 ^f,g^ ± 0.20
F1.10	16 October	1.50 ^g^ ± 0.08	1.12 ^g^ ± 0.07	2.89 ^g^ ± 0.15	1.04 ^g^ ± 0.04	1.82 ^g^ ± 0.09	2.86 ^g^ ± 0.13
F1.11	30 October	0.60 ^h^ ± 0.04	0.52 ^h^ ± 0.03	1.29 ^h^ ± 0.07	1.00 ^g^ ± 0.03	1.56 ^h^ ± 0.06	2.56 ^g^ ± 0.09
F2.1	29 May	ND	ND	ND	8.02 ^a^ ± 0.25	8.26 ^a^ ± 0.28	16.27 ^a^ ± 0.53
F2.2	12 June	ND	ND	ND	6.12 ^b^ ± 0.26	8.53 ^a^ ± 0.12	14.65 ^b^ ± 0.38
F2.3	26 June	ND	ND	ND	6.73 ^b^ ± 0.72	5.67 ^b^ ± 0.57	12.4 ^c^ ± 1.29
F2.4	10 July	ND	ND	ND	3.68 ^c^ ± 0.2	5.21 ^b^ ± 0.13	8.90 ^d^ ± 0.33
F2.5	24 July	1.62 ^a^ ± 0.19	1.38 ^a^ ± 0.23	3.16 ^a^ ± 0.42	2.55 ^d^ ± 0.2	1.91 ^c^ ± 0.2	4.45 ^e^ ± 0.40
F2.6	7 August	8.98 ^b^ ± 0.76	6.87 ^b^ ± 0.66	15.9 ^b^ ± 1.42	2.07 ^e^ ± 0.14	1.88 ^c^ ± 0.11	3.95 ^f^ ± 0.25
F2.7	21 August	11.72 ^c^ ± 0.51	6.99 ^b^ ± 0.37	18.92 ^c^ ± 0.88	1.89 ^e^ ± 0.05	1.92 ^c^ ± 0.06	3.80 ^f^ ± 0.11
F2.8	4 September	7.85 ^d^ ± 0.34	4.78 ^c^ ± 0.24	12.63 ^d^ ± 0.58	1.79 ^e^ ± 0.09	1.79 ^c^ ± 0.09	3.59 ^f^ ± 0.18
F2.9	18 September	4.66 ^e^ ± 0.33	3.76 ^d^ ± 0.27	8.66 ^e^ ± 0.60	1.63 ^e^ ± 0.07	1.98 ^c^ ± 0.09	3.61 ^f^ ± 0.16
F2.10	16 October	2.27 ^f^ ± 0.04	2.01 ^e^ ± 0.02	4.14 ^f^ ± 0.06	0.78 ^f^ ± 0.04	1.01 ^d^ ± 0.04	1.79 ^g^ ± 0.08
F2.11	30 October	3.03 ^g^ ± 0.25	2.03 ^e^ ± 0.19	5.40 ^g^ ± 0.44	1.01 ^f^ ± 0.03	1.25 ^d^ ± 0.04	2.26 ^h^ ± 0.07
F3.1	29 May	ND	ND	ND	6.22 ^a^ ± 0.25	9.26 ^a^ ± 0.28	15.48 ^a^ ± 0.53
F3.2	12 June	ND	ND	ND	6.86 ^b^ ± 0.79	7.33 ^b^ ± 0.68	14.20 ^b^ ± 1.47
F3.3	26 June	ND	ND	ND	7.68 ^c^ ± 0.26	5.90 ^c^ ± 0.48	13.58 ^c^ ± 0.74
F3.4	10 July	ND	ND	ND	5.75 ^d^ ± 0.37	5.52 ^c^ ± 0.37	11.27 ^d^ ± 0.74
F3.5	24 July	0.46 ^a^ ± 0.04	0.50 ^a^ ± 0.00	1.04 ^a^ ± 0.04	2.19 ^e^ ± 0.16	1.98 ^d^ ± 0.14	4.17 ^e^ ± 0.30
F3.6	7 August	7.24 ^b^ ± 0.29	5.87 ^b^ ± 0.23	13.13 ^b^ ± 0.52	1.4 ^f^ ± 0.04	2.03 ^d^ ± 0.06	3.43 ^f^ ± 0.10
F3.7	21 August	5.34 ^c^ ± 0.4	3.93 ^c^ ± 0.29	9.85 ^c^ ± 0.69	1.12 ^f^ ± 0.11	1.76 ^d^ ± 0.17	2.88 ^g^ ± 0.28
F3.8	4 September	5.25 ^c^ ± 0.39	4.35 ^d^ ± 0.37	9.6 ^c^ ± 0.76	1.22 ^f^ ± 0.12	1.99 ^d^ ± 0.16	3.21 ^f,g^ ± 0.28
F3.9	18 September	3.99 ^d^ ± 0.01	3.43 ^c^ ± 0.05	7.54 ^d^ ± 0.06	1.12 ^f^ ± 0.09	1.91 ^d^ ± 0.14	3.03 ^g^ ± 0.23
F3.10	16 October	2.01 ^e^ ± 0.06	1.21 ^e^ ± 0.04	3.10 ^e^ ± 0.10	0.71 ^g^ ± 0.06	1.19 ^e^ ± 0.08	1.90 ^h^ ± 0.14
	2017						
F1.1	29 May	ND	ND	ND	7.59 ^a^ ± 0.62	7.87 ^a^ ± 0.72	15.46 ^a^ ± 1.34
F1.2	12 June	ND	ND	ND	8.16 _b_ ± 0.08	6.81 ^b^ ± 0.05	14.97 ^b^ ± 0.13
F1.3	26 June	ND	ND	ND	6.79 ^c^ ± 0.14	5.23 ^c^ ± 0.13	12.02 ^c^ ± 0.27
F1.4	10 July	0.12 ^a^ ± 0.02	0.05 ^a^ ± 0.01	0.18 ^a^ ± 0.03	5.58 ^d^ ± 0.01	4.64 ^d^ ± 0.01	10.22 ^d^ ± 0.02
F1.5	24 July	4.83 ^b^ ± 0.39	3.7 ^b^ ± 0.42	8.53 ^b^ ± 0.81	1.38 ^e^ ± 0.61	1.39 ^e^ ± 0.56	2.77 ^e^ ± 1.17
F1.6	7 August	7.93 ^c^ ± 1.12	6.46 ^c^ ± 0.67	14.38 ^c^ ± 1.79	1.53 ^e^ ± 0.20	1.63 ^e^ ± 0.23	3.16 ^f^ ± 0.43
F1.7	4 September	7.50 ^c^ ± 0.32	6.13 ^c^ ± 0.48	13.6 ^c^ ± 0.80	1.32 ^e^ ± 0.06	1.77 ^e^ ± 0.08	3.09 ^f^ ± 0.14
F1.8	18 September	2.16 ^d^ ± 0.12	1.81 ^d^ ± 0.10	3.97 ^d^ ± 0.22	1.25 ^e^ ± 0.08	1.52 ^e^ ± 0.12	2.77 ^g^ ± 0.20
F1.9	2 October	1.35 ^e^ ± 0.24	1.35 ^e^ ± 0.15	2.7 ^e^ ± 0.39	0.58 ^f^ ± 0.12	0.91 ^f^ ± 0.16	1.50 ^h^ ± 0.28
F1.10	16 October	1.23 ^e^ ± 0.08	1.12 ^e^ ± 0.07	2.35 ^e^ ± 0.15	0.55 ^f^ ± 0.03	1.06 ^f^ ± 0.06	1.61 ^h^ ± 0.09
F2.1	12 June	ND	ND	ND	9.72 ^a^ ± 1.12	7.48 ^a^ ± 0.36	17.20 ^a^ ± 1.48
F2.2	26 June	ND	ND	ND	9.03 ^b^ ± 0.82	6.2 ^b^ ± 0.38	15.23 ^b^ ± 1.20
F2.3	10 July	ND	ND	ND	8.12 ^c^ ± 0.41	5.43 ^c^ ± 0.25	13.55 ^c^ ± 0.66
F2.4	24 July	0.85 ^a^ ± 0.07	0.54 ^a^ ± 0.04	1.44 ^a^ ± 0.11	4.95 ^d^ ± 0.44	3.82 ^d^ ± 0.33	8.77 ^d^ ± 0.77
F2.5	7 August	3.30 ^b^ ± 0.12	2.23 ^b^ ± 0.09	5.88 ^b^ ± 0.21	2.00 ^e^ ± 0.10	1.72 ^e^ ± 0.09	3.72 ^e^ ± 0.19
F2.6	21 August	5.46 ^c^ ± 0.2	3.87 ^c^ ± 0.07	9.6 ^c^ ± 0.27	1.43 ^f^ ± 0.06	1.92 ^e^ ± 0.07	3.35 ^e^ ± 0.13
F2.7	4 September	5.65 ^c^ ± 0.45	3.89 ^c^ ± 0.38	9.56 ^c^ ± 0.83	1.7 ^e,f^ ± 0.12	1.72 ^e^ ± 0.14	3.42 ^e^ ± 0.26
F2.8	18 September	5.78 ^d^ ± 0.23	4.45 ^d^ ± 0.15	10.65 ^d^ ± 0.38	1.85 ^e,f^ ± 0.08	1.81 ^e^ ± 0.07	3.65 ^e^ ± 0.15
F2.9	2 October	5.81 ^d^ ± 0.20	4.24 ^d^ ± 0.50	10.22 ^d^ ± 0.69	1.79 ^e,f^ ± 0.4	1.73 ^e^ ± 0.42	3.51 ^e^ ± 0.82
F2.10	16 October	4.37 ^e^ ± 0.24	3.96 ^c^ ± 0.19	8.3 ^e^ ± 0.43	1.73 ^e,f^ ± 0.07	1.83 ^e^ ± 0.08	3.56 ^e^ ± 0.15
F3.1	29 May	ND	ND	ND	7.99 ^a^ ± 0.82	10.33 ^a^ ± 1.44	18.32 ^a^ ± 2.26
F3.2	12 June	ND	ND	ND	7.86 ^a^ ± 1.91	7.84 ^b^ ± 1.52	15.70 ^b^ ± 3.43
F3.3	26 June	ND	ND	ND	8.03 ^a^ ± 1.5	5.74 ^c^ ± 1.10	13.78 ^c^ ± 2.6
F3.4	10 July	ND	ND	ND	4.45 ^b^ ± 1.29	3.44 ^d^ ± 0.84	7.88 ^d^ ± 2.13
F3.5	24 July	2.36 ^a^ ± 0.13	1.58 ^a^ ± 0.1	4.18 ^a^ ± 0.23	1.67 ^c^ ± 0.13	1.53 ^e^ ± 0.13	3.2 ^e^ ± 0.26
F3.6	7 August	3.15 ^b^ ± 0.30	1.98 ^b^ ± 0.21	5.25 ^b^ ± 0.51	1.48 ^c^ ± 0.09	1.64 ^e^ ± 0.12	3.13 ^e^ ± 0.21
F3.7	21 August	5.99 ^c^ ± 0.59	3.73 ^c^ ± 0.37	10.53 ^c^ ± 0.96	1.41 ^c^ ± 0.11	1.67 ^e^ ± 0.12	3.08 ^e^ ± 0.23
F3.8	4 September	5.88 ^d^ ± 0.53	3.72 ^c^ ± 0.35	9.60 ^d^ ± 0.88	1.1 ^d^ ± 0.07	1.59 ^e^ ± 0.10	2.69 ^e^ ± 0.17
F3.9	18 September	4.73 ^e^ ± 0.25	3.58 ^d^ ± 0.18	8.31 ^e^ ± 0.43	1.26 ^c^ ± 0.09	1.56 ^e^ ± 0.10	2.82 ^e,f^ ± 0.19
F3.10	2 October	4.61 ^e^ ± 0.27	2.93 ^e^ ± 0.22	8.00 ^e^ ± 0.49	1.48 ^c,d^ ± 0.11	1.86 ^e^ ± 0.14	3.34 ^e^ ± 0.25
F3.11	16 October	2.69 ^f^ ± 0.15	1.61 ^a^ ± 0.08	4.66 ^f^ ± 0.23	0.99 ^d^ ± 0.08	1.28 ^f^ ± 0.10	2.26 ^f^ ± 0.18

**Table 3 antioxidants-13-00462-t003:** The antioxidant properties and total content levels of polyphenols and flavonoids of chokeberry fruits, expressed per 1 g of dry weight DW; different letters mean statistical difference by Tukey’s test (*p* < 0.05); Farm 1 (F1.x), Farm 2 (F2.x), Farm 3 (F3.x).

Sample No.	Date of Collection	DPPH [TE, umol/g]	FRAP [TE, umol/g]	Total Polyphenols [GAE mg/g]	Total Flavonoids [CE, mg/g]
	2016				
F1.1	29 May	2000 ^a^ ± 310	727 ^a^ ± 6	130 ^a^ ± 6	106 ^a^ ± 5
F1.2	12 June	1790 ^b^ ± 230	686 ^a^ ± 162	145 ^a^ ± 5	115 ^a^ ± 6
F1.3	26 June	1690 ^b^ ± 260	730 ^a^ ± 16	143 ^a^ ± 4	119 ^a^ ± 2
F1.4	10 July	1410 ^b^ ± 60	577 ^b^ ± 12	118 ^b^ ± 2	90 ^a,b^ ± 1
F1.5	24 July	346 ^c^ ± 35	181 ^c^ ± 15	32 ^c^ ± 5	25 ^c^ ± 0
F1.6	7 August	710 ^d^ ± 30	293 ^d^ ± 12	58 ^d^ ± 1	32 ^c^ ± 1
F1.7	21 August	550 ^e,c^ ± 50	732 ^a^ ± 18	62 ^d^ ± 1	36 ^c^ ± 0
F1.8	4 September	532 ^e,c^ ± 25	242 ^d^ ± 12	54 ^d^ ± 5	26 ^d^ ± 0
F1.9	18 September	299 ^f^ ± 21	398 ^e^ ± 16	36 ^c^ ± 1	26 ^d^ ± 0
F1.10	16 October	320 ^f^ ± 24	378 ^e^ ± 16	33 ^c^ ± 3	25 ^d^ ± 0
F1.11	30 October	216 ^g^ ± 32	254 ^d^ ± 5	26 ^e^ ± 4	18 ^e^ ± 1
F2.1	29 May	1000 ^a^ ± 100	374 ^a^ ± 20	95 ^a^ ± 7	72 ^a^ ± 7
F2.2	12 June	1900 ^b^ ± 130	696 ^b^ ± 44	140 ^b^ ± 2	106 ^b^ ± 4
F2.3	26 June	2020 ^b^ ± 80	717 ^b^ ± 9	151 ^c^ ± 6	119 ^b^ ± 2
F2.4	10 July	2070 ^b^ ± 210	693 ^b^ ± 6	148 ^c^ ± 6	119 ^b^ ± 2
F2.5	24 July	288 ^c^ ± 43	160 ^c^ ± 15	19 ^d^ ± 1	19 ^c^ ± 0
F2.6	7 August	575 ^d^ ± 72	325 ^a^ ± 10	55 ^e^ ± 4	33 ^d^ ± 1
F2.7	21 August	596 ^d^ ± 82	676 ^b^ ± 31	56 ^e^ ± 2	37 ^d^ ± 1
F2.8	4 September	720 ^e^ ± 35	359 ^a^ ± 15	58 ^e^ ± 1	33 ^d^ ± 1
F2.9	18 September	439 ^f^ ± 19	524 ^d^ ± 43	50 ^e^ ± 1	31 ^d^ ± 1
F2.10	16 Octoctober	261 ^c^ ± 24	345 ^a^ ± 13	34 ^f^ ± 1	23 ^e^ ± 1
F2.11	30 October	367 ^g^ ± 37	452 ^e^ ± 26	38 ^f^ ± 2	27 ^e^ ± 2
F3.1	29 May	1650 ^a^ ± 80	589 ^a^ ± 7	126 ^a^ ± 4	110 ^a^ ± 1
F3.2	12 June	1380 ^b^ ± 120	631 ^a^ ± 21	134 ^b^ ± 3	99 ^b^ ± 3
F3.3	26 June	1634 ^a^ ± 38	835 ^b^ ± 26	160 ^c^ ± 7	119 ^a^ ± 2
F3.4	10 July	1680 ^a^ ± 80	664 ^a^ ± 27	126 ^a^ ± 5	96 ^b^ ± 2
F3.5	24 July	369 ^c^ ± 23	270 ^c^ ± 7	40 ^d^ ± 2	26 ^c^ ± 2
F3.6	7 August	420 ^c^ ± 37	587 ^a^ ± 22	48 ^d^ ± 2	34 ^d^ ± 3
F3.7	21 August	432 ^c^ ± 37	602 ^a^ ± 18	49 ^d^ ± 4	33 ^d^ ± 1
F3.8	4 September	486 ^d^ ± 43	709 ^d^ ± 29	60 ^e^ ± 5	44 ^e^ ± 3
F3.9	18 Septembr	335 ^c^ ± 26	386 ^e^ ± 25	45 ^d^ ± 3	27 ^c^ ± 1
F3.10	16 October	362 ^c^ ± 17	460 ^f^ ± 23	45 ^d^ ± 3	32 ^d^ ± 2
	2017				
F1.1	29 May	2170 ^a^ ± 120	563 ^a^ ± 3	121 ^a^ ± 1	102 ^a^ ± 3
F1.2	12 June	1960 ^a^ ± 90	585 ^a^ ± 4	119 ^a^ ± 3	108 ^a^ ± 1
F1.3	26 June	1920 ^a^ ± 120	589 ^a^ ± 7	126 ^a^ ± 4	110 ^a^ ± 1
F1.4	10 July	1630 ^b^ ± 90	464 ^b^ ± 11	120 ^a^ ± 1	103 ^a^ ± 2
F1.5	24 July	594 ^c^ ± 52	312 ^c^ ± 23	35 ^b^ ± 5	19 ^b^ ± 3
F1.6	7 August	533 ^c^ ± 21	399 ^d^ ± 3	46 ^c^ ± 2	24 ^b^ ± 0
F1.7	4 September	566 ^c^ ± 24	442 ^b^ ± 1	51 ^d^ ± 3	24 ^b^ ± 1
F1.8	18 September	340 ^d^ ± 21	387 ^d^ ± 5	42 ^c^ ± 2	20 ^b^ ± 1
F1.9	2 October	482 ^e^ ± 46	200 ^e^ ± 5	26 ^e^ ± 1	15 ^c^ ± 0
F1.10	16 October	463 ^e^ ± 25	184 ^e^ ± 7	25 ^e^ ± 1	15 ^c^ ± 1
F2.1	12 June	1880 ^a^ ± 90	659 ^a^ ± 7	143 ^a^ ± 2	120 ^a^ ± 1
F2.2	26 June	1810 ^a^ ± 51	669 ^a^ ± 3	149 ^a^ ± 1	119 ^a^ ± 2
F2.3	10 July	1790 ^a^ ± 63	672 ^a^ ± 7	147 ^a^ ± 2	115 ^a^ ± 1
F2.4	24 July	1330 ^b^ ± 90	395 ^b^ ± 11	88 ^b^ ± 2	69 ^b^ ± 2
F2.5	7 August	440 ^c^ ± 26	209 ^c^ ± 1	33 ^c^ ± 1	17 ^c^ ± 1
F2.6	21 August	475 ^c^ ± 44	283 ^d^ ± 5	37 ^d^ ± 0	20 ^c^ ± 1
F2.7	4 September	360 ^d^ ± 30	266 ^d^ ± 2	32 ^c^ ± 1	17 ^c^ ± 0
F2.8	18 September	511 ^e^ ± 18	357 ^e^ ± 18	48 ^e^ ± 3	27 ^d^ ± 0
F2.9	2 October	544 ^e^ ± 51	383 ^b^ ± 8	47 ^e^ ± 1	27 ^d^ ± 0
F2.10	16 October	474 ^f^ ± 15	338 ^e^ ± 8	42 ^e^ ± 0	25 ^d^ ± 1
F3.1	29 May	1700 ^a^ ± 150	432 ^a^ ± 9	118 ^a^ ± 3	85 ^a^ ± 3
F3.2	12 June	2158 ^b^ ± 45	548 ^b^ ± 29	144 ^b^ ± 3	125 ^b^ ± 3
F3.3	26 June	2116 ^b^ ± 91	657 ^c^ ± 11	150 ^b^ ± 2	121 ^b^ ± 1
F3.4	10 July	1780 ^c^ ± 110	508 ^d^ ± 15	122 ^a^ ± 3	121 ^b^ ± 2
F3.5	24 July	417 ^d^ ± 22	209 ^e^ ± 10	32 ^c^ ± 1	18 ^c^ ± 1
F3.6	7 August	580 ^e^ ± 50	331 ^f^ ± 10	45 ^d^ ± 1	29 ^d^ ± 1
F3.7	21 August	644 ^f^ ± 22	386 ^g^ ± 5	51 ^d^ ± 0	28 ^d^ ± 0
F3.8	4 September	561 ^g^ ± 18	387 ^g^ ± 4	50 ^d^ ± 2	27 ^d^ ± 0
F3.9	18 September	594 ^g^ ± 7	372 ^g^ ± 8	49 ^d^ ± 0	28 ^d^ ± 1
F3.10	2 October	565 ^g^ ± 12	383 ^g^ ± 15	52 ^d^ ± 0	30 ^d^ ± 1
F3.11	16 October	526 ^h^ ± 25	322 ^f^ ± 16	43 ^e^ ± 1	25 ^d^ ± 1

## Data Availability

Data is contained within the article.

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
