# Peer review of "Profile of Bioactive Components and Antioxidant Activity of Aronia melanocarpa Fruits at Various Stages of Their Growth, Using Chemometric Methods"

_antioxidants, 2024, doi:10.3390/antiox13040462_

Round 1
Reviewer 1 Report
The manuscript brings some useful information regarding the correct utilization of the fruits of Aronia melanocarpa as raw material for obtaining functional food and dietary supplements
The manuscript presents an interesting study on the chemical composition and the antioxidant activity of Aronia melanocarpa fruits, harvested in different periods of development.
In order to improve the quality of the manuscript, I have some observations and recommendations as follows:
- The scientific name of the species – Aronia melanocarpa – should be written in italic throughout the text
- 3.1. HPLC Analysis of Chlorogenic Acids and Anthocyanins - Line 193 - Figure 1 instead of 2; line 195 – Figure 1a instead of 2a
- It is not very clear which were the analyzed chlorogenic acids and anthocianosides. More details concerning the characteristics of the identified compounds are needed. In 2.8. HPLC analysis – are presented the Retention times values for nCGA, CGA, iCGA, CyaGal – meaning that these were the only reference compounds available? In Table S1 appear Cya Ara, CyaGal, nCGA, CGA - it needs accordance. The abbreviations must be also explained.
- Flavonoids are the major polyphenolic compounds, why were not analyzed and identified?
- The CGAs and anthocyanin contents – figure 1 – refer to quantitative determinations described in 2.3, 2.4, or they are the results of HPLC analysis?
- In tables S1, S2, the name of the samples is not explained – LS1, ZG…….
- I consider useful to present the results of the determinations of Total anthocyanin content, Total phenolic content, Total flavonoid content (Table S2) in chapter 3. Results and discussion, instead of supplementary material, and the same for table S1.
Reviewer 2 Report
First, more extensive size experiments (locations 3 * sampling 11 * years 2) are impressive.
However, the manuscript should be significantly improved before publication, particularly in the Results and discussion.
M&M
Table 1 is not informative. It can be described as a few sentences. Instead, I would recommend making a table on farms (environments) and harvest samples (color of fruits, etc.).
RESULTS AND DISCUSSION
This part should be improved logically and comprehensively.
1) The resonance of figures should be improved. The dates in Fig. 1 and 3 do not correspond to the date in Table 1.
2) What is the significance (importance) of Ara/Gal and nCGA/CGA in Fig. 3? It should be addressed in the Discussion.
3) What are the criteria for mid-July? colors? maturation stage? What’s the regular harvest season?
Yang et al.(2019) reported that the red tip stage of Aronia fruits contains more flavonoids, phenolic compounds, and polyphenols than the dark purple stage, with higher antioxidant activity. So, does before mid-July correspond to the red tip stage?
The August 13 – September 2 period showed the highest CGAs in Figure 1. Does that mean this stage is the harvest season on farms?
4) Usually, the Discussion may be separated from the Results in high-quality articles. And this part should improved.
5) The shifting trend of polyphenolics and anthocyanin during the ripening of fruits is typical. Hence, more discussions are recommended.
6) Any discussion on the effect of farms can be addressed in the Discussion, in addition to the effect of harvest season.
Reviewer 3 Report
This manuscript improves the field of knowledge of phytochemical compounds and bioactive properties of Aronia melanocarpa fruits by revealing new findings on their bioactive components and antioxidant activity.
The information obtained has the potential to advance the understanding of the specific bioactive compounds primarily responsible for the antioxidant properties of chokeberry.
The paper also provides significant information, in line with the current state of knowledge in the field, regarding the impact of geographical region on the antioxidant properties of Aronia berries and their variability during ripening. The results are helpful to characterize and classify Aronia samples during growth according to their geographical origin. Thus, this study contributes significantly to this field and are of major help when the unripe Aronia berries are considered in the design of functional food ingredients and dietary supplements in the future, similar to ripe A. melanocarpa fruits.
The purpose of the paper is clearly stated and the literature review has critically followed.
The issues addressed are relevant to the field and aim to fill some of the existing gaps, as no studies have focused on the biochemical composition and biological activity of Aronia samples from different regions of Poland. Please add in the Introduction, 1-2 more sentences in which you point out the shortcomings in the field, to better motivate this study.
The research methodology is adequate, well described and sufficiently detailed.
The results were clearly reported in 5 figures and discussed appropriately. In figures 1, 2 and 4, the two parts a and b should be enlarged more and rearranged vertically (one below the other) for clarity.
Given that this approach has great potential to be used for the preparation of other plant-based products, please expand the discussion by highlighting the innovative elements brought by this research.
In general, the conclusions are consistent with the evidence and arguments presented, but need to be improved to sufficiently answer the main question addressed. In addition, the authors have not sufficient highlighted the added value of their work, this should be added in the conclusion.
The references are relevant to this research topic.
The content of this manuscript fits well with the topic of the journal, makes an important contribution to the field and could be published in Antioxidants after revision.
Round 2
Reviewer 1 Report
The authors improved the quality of the manuscript and they answered to the observations and recommendations of the reviewers.
The authors completed the manuscript according to the reviewers’ recommendations
Reviewer 2 Report
The manuscript was significantly improved. Well done.
.